# PARAMAGENT: LANGUAGE AGENTS WITH PARAMETRIC KNOWLEDGE

## ABSTRACT

Large Language Models (LLMs) have demonstrated strong reasoning abilities, yet existing agent frameworks remain constrained by two limitations. First, they typically operate at the per-instance level, confining signals to individual problems and overlooking transferable patterns across tasks. Second, while some approaches attempt to incorporate global information through external memory, these are non-parametric in nature, and thus capture only shallow interactions across instances, failing to uncover deeper regularities. To overcome these limitations, we propose `ParamAgent`, a language agent framework that leverages a domain-adaptive parametric module to internalize knowledge across samples into model parameters. In addition to capturing cross-sample regularities, `ParamAgent` provides twofold flexibility: (i) the parametric module can supply different forms of knowledge depending on various domains, and (ii) the same module can be integrated with different base LLMs, making `ParamAgent` broadly applicable. Moreover, `ParamAgent` naturally promotes diversity of outputs by adjusting the sampling temperature of the parametric module. Experiments on programming, math reasoning, and multi-hop question answering benchmarks show that `ParamAgent` consistently outperforms state-of-the-art baselines, surpassing the best baseline by up to 7.90%, 9.41%, and 24.30% respectively.

## 1 INTRODUCTION

Large language models (LLMs) (Brown et al., 2020; Chowdhery et al., 2023; Touvron et al., 2023) have exhibited striking progress in complex reasoning tasks. Their ability to interleave reasoning with actions has led to the development of autonomous language agents that treat an LLM as the core policy(Wei et al., 2022; Shinn et al., 2023; Yao et al., 2023b;a; Zhou et al., 2023; Madaan et al., 2023; Wu et al., 2023; Wang et al., 2023a; Hong et al., 2024; Zhang et al., 2024a). For example, Chain-of-Thought (CoT) prompting (Wei et al., 2022) elicits explicit intermediate steps that improve reasoning performance on complex tasks. Self-Refine (Madaan et al., 2023) introduces an iterative self-feedback loop, enabling models to progressively refine their outputs and achieve higher-quality results. Subsequent work expands agents' search and feedback mechanisms: Reflexion (Shinn et al., 2023) stores verbalized feedback in episodic memory (i.e., a long-term memory of the agent's self-reflections accumulated across iterations) and yields noticable gains; Tree-of-Thoughts (ToT) (Yao et al., 2023a) explores multiple reasoning paths via tree search; LATS integrates Monte-Carlo Tree Search for long-horizon planning (Zhou et al., 2023; Browne et al., 2012).

Despite the effectiveness of self-reflection, recent work has identified a lack of diversity in the reflective signals as the limitation (Lingam et al., 2025). To address this, Lingam et al. (2025) has proposed DoT and DoT-bank, which enhance reflective diversity through prompt-level variation and retrieval-based cross-sample trajectories. Similarly, many previous work propose to use textual log as external memory to enrich the reflective inputs, therefore enhancing the reasoning ability of language agent (Borgeaud et al., 2022; Shi et al., 2023; Wang et al., 2023c; Shi et al., 2023; Zheng et al., 2023; Wang et al., 2024). These results confirm that introducing diverse reflective information can substantially improve the agent's reasoning process. However, prompt-based approaches cannot capture cross-sample regularities, and retrieval-based methods using external memory may be constrained by the limited number and coverage of stored samples. This naturally raises the question: **How can we further expand reflective diversity to achieve stronger reasoning performance?**

To overcome these limitations, we propose `ParamAgent`, a language-agent framework that leverages parametric knowledge. `ParamAgent` introduces an external module $M_*$ (where $*$ denotes different knowledge types across domains) to internalize cross-sample information. When solving a new problem, the agent queries $M_*$ to obtain population-level insights rather than purely instance-level feedback. For example, $M_r$ can synthesize common error patterns for programming and math tasks, while $M_p$ generates semantic decomposition units for multi-hop question answering (Sec. 3). As the parametric knowledge is encoded directly into the parameters of $M_*$ through training, the module captures cross-sample regularities and produces reflective signals that differ fundamentally from self-reflection and retrieval-based trajectories. This parametric module introduces an additional layer of diversity into the reflective inputs. As we will show in Sec. 4, this additional form of diversity works jointly with reflection-based frameworks and further enhances the agent's reasoning ability. Beyond diversity, `ParamAgent` offers two forms of flexibility. First, $M_*$ can generate different types of knowledge, from population-level reflective feedback to structured semantic units. Second, the same parametric module can be paired with different base LLMs, making `ParamAgent` broadly applicable across architectures and domains.

We evaluate `ParamAgent` on math reasoning problems, programming, and multi-hop QA. In each domain, `ParamAgent` significantly outperforms state-of-the-art methods. Concretely, our approach surpasses the best baseline by up to 7.90% on programming, 9.41% on math reasoning, and 24.30% on multi-hop QA. Our contributions can be summarized as follows:

- We identify key limitations in existing agent frameworks and propose leveraging parametric knowledge to capture cross-sample interactions, thereby augmenting the reasoning process.

- We propose `ParamAgent`, a language agent that equips a parametric module to capture cross-sample regularities, and further introduce `ParamAgent-plus`, an enhanced variant that integrates multiple forms of memory modules.

- The parametric module $M_*$ is capable of synthesizing multiple forms of knowledge that support adaptation to a wide range of domains, and it can be flexibly integrated with different base LLMs in the agents.

- Through extensive experiments on programming, math reasoning, and multi-hop QA, `ParamAgent` consistently outperforms state-of-the-art baselines, surpassing the second best by up to 7.90%, 9.41%, and 24.30% respectively.

## 2 PRELIMINARIES

We consider a pretrained Language Model (LM) $p_\theta$ with parameters $\theta$ that operates on token sequences. Let $x = (x[1], \ldots, x[l_x])$ denote the input sequence and $y = (y[1], \ldots, y[l_y])$ the output sequence. The LM decodes autoregressively, i.e., $p_\theta(y \mid x) = \prod_{i=1}^{l_y} p_\theta(y[i] \mid x, y[1{:}i{-}1])$, and, more generally, with an auxiliary prompt $\pi$ (e.g., instructions, exemplars, tool feedback, etc.), $p_\theta(y \mid x, \pi) = \prod_{i=1}^{l_y} p_\theta(y[i] \mid x, \pi, y[1{:}i{-}1])$. We use $z_1, \ldots, z_n$ to denote intermediate thoughts, and $r_1, \ldots, r_k$ to denote self-reflections. A node in a search tree is written as $s = [x, z_{1:i}]$.

**Input–Output (IO) prompting** The LM is prompted with task instructions and/or few-shot IO pairs and directly produces the final output: $y \sim p_\theta(\cdot \mid x, \pi_{\text{IO}})$.

**Chain-of-Thought (CoT)** To handle $x \mapsto y$, CoT (Wei et al., 2022) instructs the model to first generate a sequence of thoughts and then the answer: $z_i \sim p_\theta(\cdot \mid x, z_{1:i-1})$, $y \sim p_\theta(\cdot \mid x, z_{1:n})$. In practice, $[z_{1:n}, y]$ is sampled as a single contiguous sequence.

**Reflexion** Reflexion (Shinn et al., 2023) augments the prompt with episodic self-reflections $r_{1:k}$ from the previous $k$ iterations. The agent then generates new solutions conditioned on the previous feedbacks: $y \sim p_\theta(\cdot \mid x, r_{1:k})$. Intuitively, $r_i$ provides textual semantic gradient signals (Shinn et al., 2023; Yuksekgonul et al., 2024), indicating common errors to avoid and corrective cues.

**Diversity of Thoughts (DoT)** DoT (Lingam et al., 2025) enhances the diversity of reflection feedback by using explicit prompt-level instructions to generate a set of diversified reflections

(a) An output example of $M_r$ on programming task using HumanEval dataset.

(b) An output example of $M_p$ on multi-hop QA task using HotpotQA dataset.

Figure 1: Illustration of the output produced by $M_*$

$\{r_i\}_{i=1}^k$, thereby reducing redundancy and improving coverage of solutions. The decoding objective remains: $y \sim p_\theta(\cdot \mid x, r_{1:k})$.

**Tree-of-Thought (ToT) and LATS** ToT (Yao et al., 2023a)lifts CoT into a search over partial solutions $s = [x, z_{1:i}]$. New thoughts are proposed via CoT-style sampling $z_i \sim p_\theta(\cdot \mid x, z_{1:i-1})$, while DFS/BFS is used to explore the search tree. LATS (Zhou et al., 2023) extends this view with Monte-Carlo Tree Search (MCTS) (Browne et al., 2012), repeatedly selecting, expanding, simulating, and backpropagating values over nodes $s$, thereby constructing high-value trajectories of thoughts leading to a more probable correct $y$.

## 3 AUGMENTING LANGUAGE AGENTS WITH PARAMETRIC KNOWLEDGE

In this section, we show how `ParamAgent` leverages parametric knowledge to augment LLM-based agents. Depending on the domain, `ParamAgent` employs different forms of parametric knowledge. Speficically, **(1)** A reflection-oriented module $M_r$ to synthesize model-based reflection for programming and math, and **(2)** A decomposition-oriented module $M_p$ to produce semantic units for multi-hop QA. The detailed pseudo-code can be found in Appendix B, a shorter version can be found in Algorithm 1.

### 3.1 GLOBAL-LOCAL REFLECTION

To incorporate cross-sample reflective signals beyond instance-specific cues, we propose a *global–local reflection* mechanism. The key idea is to combine self-reflections derived from episodic memory with global reflections synthesized by a parametric module $M_r$.

**Training $M_r$.** We obtain $M_r$ by fine-tuning a pretrained LLM on a curated dataset where reflective feedback is provided as supervision. Through this process, the module internalizes population-level patterns into its model parameters, and learns to synthesize model-based reflections and corrective cues. We provide more details regarding the dataset curation and training in Appendix D.2. An example output from $M_r$ is shown in Figure 1a.

**Formulation.** Having obtained $M_r$, `ParamAgent` conditions jointly on two sources of feedback:

$$y \sim p_\theta(\cdot \mid x, r_{1:k}, r_k^g), \quad r_k^g \sim p_\psi(\cdot \mid x), \tag{1}$$

where $r_{1:k}$ denotes $k$ self-reflections collected across iterations, and $r_k^g$ is the global reflection generated by $M_r$ at the $k$-th iteration.

**Usage.** In each episode, $M_r$ samples a global reflection $r_k^g$, which is injected into the prompt alongside self-reflections from the memory. A low sampling temperature is used in the first round to ensure informative feedback, while later rounds adopt a higher temperature to promote diversity.

Importantly, $r_k^g$ is used only as a transient input and is not stored in memory; the episodic memory only maintains self-reflections generated by the agent itself.

## 3.2 SEMANTIC DECOMPOSITION

Beyond reflections, the parametric module can also generate structured knowledge. Inspired by *chunking* and the *working memory* model from cognitive science (Miller, 1956; Baddeley, 2020), we introduce $M_p$ to decompose complex multi-hop queries into compact semantic units that guide reasoning, one such example is illustrated in Figure 1b.

**Training $M_p$.** Similar to $M_r$, we fine-tune a pretrained LLM where semantic decompositions (e.g., entities, relations, constraints, answer types, etc.) serve as the training signal. Details are deferred in Appendix D.2.

**Formulation.** `ParamAgent` then conditions jointly on the original query $x$, the semantic units $Z$, and a set of self-reflections $r_{1:k}$ derived from prior attempts, with the final answer produced as:

$$y \sim p_\theta(\cdot \mid x, Z, r_{1:k}). \qquad (2)$$

---

**Algorithm 1:** Pseudocode for `ParamAgent`

**Require:** Dataset $\mathcal{D}$, base LM $p_\theta$, parametric module $M_*$ with params $\psi$, max iterations $T_{\max}$
1: $\mathcal{M} \leftarrow \emptyset$         ▷ Initialize memory
2: **for** $x \in \mathcal{D}$ **do**
3:     **for** $t = 1$ to $T_{\max}$ **do**
4:        $T \leftarrow \begin{cases} 0.2 & \text{if } t = 1 \\ 1.0 & \text{otherwise} \end{cases}$
5:        $G_{t-1} \sim p_\psi(\cdot \mid x; T)$    ▷ $r_{t-1}^g$ **or** $Z$
6:        $r_{1:t-1} \leftarrow$ RETRIEVEREFLECTIONS$(\mathcal{M}, x)$
7:        $y_t \sim p_\theta(\cdot \mid x, r_{1:t-1}, G_{t-1})$
8:        **if** EVALUATE$(y_t, x)$ **then**
9:           break
10:       **else**
11:          $r_t \leftarrow$ GENERATESELFREFLECTION$(y_t)$
12:          $\mathcal{M} \leftarrow \mathcal{M} \cup \{(x, r_t)\}$    ▷ **Store reflection**
13:       **end if**
14:     **end for**
15: **end for**

---

By combining self reflection with model-based semantic decomposition, the agent benefits from both local reflective feedback and global structural guidance.

**Usage.** Similarly, at the first round, $M_p$ generates semantic units under a low temperature, and the temperature is increased in the remaining rounds to promote diversity.

## 3.3 RELATION TO PREVIOUS STUDIES

**Global-local reflection** Our design is inspired by Reflexion (Shinn et al., 2023), which improves reasoning through iterative interaction between an actor $M_a$, an evaluator $M_e$, and a self-reflection module $M_{sr}$. The actor generates candidate outputs, the evaluator provides feedback based on task-specific signals, and the self-reflection module produces natural language feedback that is appended to the context of the next trial. In `ParamAgent`, we propose a new module $M_r$, which encodes cross-sample similarities into the model parameters, allowing the module to generate higher-quality reflections that facilitate the reasoning process of the language agent.

**Semantic decomposition** A key advantage of `ParamAgent` lies in its flexibility: the method can provide different forms of knowledge depending on the task. For multi-hop QA, we introduce semantic decomposition, which is conceptually related to CoT prompting. In standard CoT, intermediate thoughts $z_i$ are elicited directly by the base LM. In contrast, our framework employs a dedicated parametric module $M_p$, which generates a structured set of intermediate thoughts $Z$ that guide the reasoning process. Furthermore, we show that semantic decomposition can be seamlessly combined with reflection-based framework. Together they complement each other, yielding richer guidance to augment reasoning.

## 4 EXPERIMENTS

In this section, we detail our experimental setup and present results across programming, math reasoning, and multi-hop QA. We then conduct more in-depth empirical analyses of our proposed

method. More experimental results are included in Appendix D, including experiments with 70B scale LLMs.

## 4.1 SETUP

**Datasets** We evaluate our framework across three domains. For programming, we use HumanEval (Chen et al., 2021) and MBPP (Austin et al., 2021). For math reasoning, we adopt the MATH dataset (Hendrycks et al., 2021b), which covers competition-level problems of varying difficulty across seven subjects. For multi-hop QA, we use HotpotQA (Yang et al., 2018) and 2WikiMultiHopQA (Ho et al., 2020), which require reasoning across multiple passages. Further details about each dataset, as well as how we perform dataset splits are provided in later sections.

**Evaluation** For programming tasks, we follow prior work (Shinn et al., 2023; Lingam et al., 2025) and report Pass@1. During generation, only visible or synthetic test cases are used, while final evaluation is conducted on hidden test cases; a score of 1 is assigned if all tests pass and 0 otherwise. For math reasoning and multi-hop QA, we report 0–1 accuracy on subsampled testsets.

**Baselines** We compare against: (1) **Base**, the underlying LLM agent without reflection; (2) **Reflexion** (Shinn et al., 2023), which uses episodic self-reflections; (3) **DoT** (Lingam et al., 2025), which augments Reflexion with prompt-level diversity. (4) **DoT-bank** (Lingam et al., 2025), which further incorporate a memory bank to enrich the reflective feedbacks. **(5)** In addition, we develop a baseline that uses only the parametric module to generate reflections or semantic units, referred to as *model-based reflection* or *model-based CoT*. This baseline utilizes $M_*$ purely as a parametric sampler, without performing self-reflection. The pseudocode can be found in Appendix B. Our full model `ParamAgent` incorporates both episodic memory and parametric memory for iterative reasoning, yielding stronger and more diverse feedback signals. Finally, we also explore an extended variant `ParamAgent-plus`, where we further introduce a memory bank similar to DoT. This extension allows the agent to combine episodic memory, parametric memory, and cross-sample memory, offering a comprehensive integration of different knowledge sources.

To ensure a comprehensive evaluation, we employ three backbone LLMs with varying levels of reasoning capability: (1) **Llama-3.1-8B** (Dubey et al., 2024), a strong open-source reasoning model; (2) **Mistral-7B-v0.2** (Jiang et al., 2023), a competitive medium-sized model with efficient inference; and (3) **Qwen2-1.5B-instruct** (Bai et al., 2023), a TogetherAI's hosted version of Qwen2 1.5B, fine-tuned into an instruction-following variant. This selection of backbones allows us to examine how our approach performs across different model sizes and reasoning strengths. We also provide results with stronger base LLMs in Appendix D.4, showing that even when the parametric module remains an 8B model, it can still provide noticable gains to agents built on 70B-scale LLMs.

**Hyperparameters** Across all experiments, we fix the number of reflection iterations to 5 for both baseline methods and our proposed approach. For `ParamAgent` and its variants, we set the sampling temperature to $T = 0.2$ during the first iteration, and $T = 1.0$ in the subsequent iterations to promote diversity. For LoRA finetuning of the parametric modules, we use a rank of $r = 128$, scaling factor $\alpha = 32$, a learning rate of $2e - 5$, and train for 3 epochs.

**Parametric module** The parametric module is designed to internalize population-level knowledge across tasks $t \in \mathcal{D}$. We first construct a dataset $\{(t_i, m_i)\}_{i=1}^n$ by prompting an LLM on synthetic data or a subset of the training set to enumerate failure modes or semantic units, where $n$ is typically around $10^4$. $M_r$ or $M_p$ is then obtained by finetuning a pretrained LLM using LoRA (Hu et al., 2022), encoding this knowledge into its parameters. In our experiments, we instantiate the module with Llama-3.1-8B. More details on the setup, dataset construction, training procedure, and implementation of the parametric module are deferred to Appendix D.2.

## 4.2 PROGRAMMING

**Datasets** We evaluate our framework on two widely used programming benchmarks: HumanEval (Chen et al., 2021) and MBPP (Austin et al., 2021). HumanEval consists of Python programming problems that test functional correctness using hidden unit tests, while MBPP covers beginner to intermediate-level Python problems designed for program synthesis.

Table 1: Performance on HumanEval and MBPP datasets. **Bold** denotes the best result, and underline marks the second best. ↑ and ↓ indicate the absolute improvement or decrease relative to the Base method. For clarity, the prompt token usage of the Base method is normalized to 1. Table 2 and Table 3 use the same notation, which we omit from the captions due to space constraints.

| Dataset | Method | Llama-3.1-8B | | Mistral-7B-v0.2 | | Qwen2-1.5B | |
|---|---|---|---|---|---|---|---|
| | | Pass@1 | #Prompt Tokens | Pass@1 | #Prompt Tokens | Pass@1 | #Prompt Tokens |
| HumanEval | Base | 59.15 | 1.00 | 32.93 | 1.00 | 41.46 | 1.00 |
| | Model-based Reflection | 78.05 ↑18.90 | 9.15 | **68.29** ↑35.36 | 23.73 | **68.91** ↑27.45 | 6.77 |
| | Reflexion | 76.22 ↑17.07 | 9.29 | 51.22 ↑18.29 | 28.54 | 49.39 ↑7.93 | 18.30 |
| | DoT | 73.17 ↑14.02 | 17.45 | 46.95 ↑14.02 | 43.06 | 56.56 ↑15.10 | 15.26 |
| | DoT-bank | 79.56 ↑20.41 | 24.71 | 54.26 ↑21.33 | 61.62 | 60.10 ↑18.64 | 31.28 |
| | Ours | **82.93** ↑23.78 | 19.18 | 67.07 ↑34.14 | 70.38 | 66.46 ↑25.00 | 33.45 |
| MBPP | Base | 47.61 | 1.00 | 24.94 | 1.00 | 42.06 | 1.00 |
| | Model-based Reflection | 52.90 ↑5.29 | 31.93 | 47.86 ↑22.92 | 20.98 | 52.89 ↑10.83 | 25.35 |
| | Reflexion | 58.69 ↑11.08 | 37.18 | 28.46 ↑3.52 | 14.02 | 47.61 ↑5.55 | 26.95 |
| | DoT | 61.21 ↑13.60 | 51.83 | 19.79 ↓5.15 | 25.45 | 47.37 ↑5.31 | 21.48 |
| | DoT-bank | 64.82 ↑17.21 | 69.41 | 24.68 ↓0.26 | 60.09 | 53.38 ↑11.32 | 60.95 |
| | Ours | **67.00** ↑19.39 | 86.39 | **51.64** ↑26.70 | 36.88 | **54.90** ↑12.84 | 66.86 |

Table 2: Performance on MATH dataset.

| Dataset | Method | Llama-3.1-8B | | Mistral-7B-v0.2 | | Qwen2-1.5B | |
|---|---|---|---|---|---|---|---|
| | | Acc | #Prompt Tokens | Acc | #Prompt Tokens | Acc | #Prompt Tokens |
| MATH | Base | 48.20 | 1.00 | 12.23 | 1.00 | 8.99 | 1.00 |
| | Model-based Reflection | 45.81 ↓2.39 | 2.58 | 13.31 ↑1.08 | 2.82 | 16.91 ↑7.92 | 2.84 |
| | Reflexion | 58.99 ↑10.79 | 23.33 | 19.78 ↑7.55 | 27.67 | 21.94 ↑12.95 | 18.39 |
| | DoT | 64.38 ↑16.18 | 34.17 | 23.25 ↑11.02 | 40.51 | 22.30 ↑13.31 | 31.99 |
| | DoT-bank | 73.02 ↑24.82 | 83.92 | 35.61 ↑23.38 | 122.92 | 24.37 ↑15.38 | 76.71 |
| | `ParamAgent` | 67.99 ↑19.79 | 57.01 | 28.06 ↑15.83 | 92.91 | 22.30 ↑13.31 | 70.07 |
| | `ParamAgent-plus` | **75.45** ↑27.25 | 111.32 | **38.96** ↑26.73 | 196.18 | **25.97** ↑16.98 | 144.25 |

**Results** From Table 1, we can observe that: **(1)** Model-based Reflection, which relies solely on parametric reflection without using self-reflection, already achieves substantial gains or performs comparably to the baseline methods across different LLM backbones. This indicates that parametric knowledge alone can provide useful reflective signals, effectively guiding the agent toward higher-quality solutions. **(2)** Furthermore, `ParamAgent`, which integrates both instance-specific self-reflections and model-based parametric reflections, achieves consistent improvements across most datasets and base models. This highlights the complementary benefits of combining feedback at different granularities: local signals capturing trial-specific errors and global signals capturing population-level patterns. **(3)** Notably, although DoT-bank also leverages global-level reflective feedback, it underperforms compared to `ParamAgent` in most scenarios. This highlights that retrieval-based memory modules are less effective than model-based parametric modules in capturing deep relational patterns across tasks. **(4)** Finally, `ParamAgent` is also cost-effective, its token consumption is on par or lower than DoT-bank while delivering stronger performance.

## 4.3 MATH REASONING

**Datasets** For mathematical reasoning, we evaluate on the MATH dataset (Hendrycks et al., 2021b), which consists of competition-level problems spanning seven subjects: Prealgebra, Algebra, Number Theory, Counting and Probability, Geometry, Intermediate Algebra, and Precalculus. To construct our evaluation set, we randomly sample 40 problems from each subject in testset, ensuring that the problems cover diverse topics and allows us to comprehensively assess the performance of different methods across varying topics.

**Results** From Table 2, we can make several observations. **(1)** Compared with model-based reflection alone, instance-level self-reflection proves more effective for math reasoning. However, when model-based reflective feedback is incorporated, `ParamAgent` consistently improves over Reflexion across all 3 backbone LLMs, and outperforms DoT on all LLM backbones. **(2)** Furthermore, `ParamAgent-plus`, which incorporates all levels of memory modules, achieves state-of-the-art results across all 3 backbone LLMs. This clearly demonstrates the added value of parametric knowledge: by combining episodic memory, parametric memory, and cross-sample memory, the agent gains access to richer and more complementary feedback signals, enabling stronger performance on competition-level mathematical reasoning tasks.

Table 3: Performance on HotpotQA and 2WikiMultiHopQA datasets.

| Dataset | Method | Llama-3.1-8B | | Mistral-7B-v0.2 | | Qwen2-1.5B | |
| | | Acc | #Prompt Tokens | Acc | #Prompt Tokens | Acc | #Prompt Tokens |
|---|---|---|---|---|---|---|---|
| HotpotQA | Base | 57.67 | 1.00 | 45.00 | 1.00 | 43.66 | 1.00 |
| | Model-based CoT | 61.67 ↑4.00 | 1.46 | 54.33 ↑9.33 | 1.46 | 48.10 ↑4.44 | 1.44 |
| | Reflexion | 71.33 ↑13.66 | 4.13 | 62.33 ↑17.33 | 4.67 | 50.03 ↑6.37 | 6.22 |
| | DoT | 66.67 ↑9.00 | 7.10 | 58.33 ↑13.33 | 8.97 | 49.32 ↑5.66 | 58.05 |
| | DoT-bank | 72.00 ↑14.33 | 13.28 | 66.33 ↑21.33 | 19.35 | 52.02 ↑8.36 | 109.54 |
| | ParamAgent | **78.33** ↑20.66 | 22.25 | **69.67** ↑24.67 | 34.99 | **64.66** ↑21.00 | 14.69 |
| 2WikiMultiHopQA | Base | 40.33 | 1.00 | 21.00 | 1.00 | 40.33 | 1.00 |
| | Model-based CoT | 54.67 ↑14.34 | 1.39 | 46.33 ↑25.33 | 1.21 | 40.66 ↑0.33 | 1.19 |
| | Reflexion | 78.67 ↑38.34 | 5.47 | 61.33 ↑40.33 | 5.86 | 51.00 ↑10.67 | 6.56 |
| | DoT | 66.67 ↑26.34 | 7.03 | 52.13 ↑31.13 | 6.40 | 47.83 ↑7.50 | 30.55 |
| | DoT-bank | 80.33 ↑40.00 | 12.49 | 74.66 ↑53.66 | 8.10 | 50.49 ↑10.16 | 54.92 |
| | ParamAgent | **88.67** ↑48.34 | 10.41 | **81.33** ↑60.33 | 14.43 | **63.33** ↑23.00 | 17.39 |

## 4.4 MULTI-HOP QA

**Datasets**   We use a subset of samples from HotpotQA and 2WikiMultiHopQA for the evaluation. Specifically, for HotpotQA, from the training set, we randomly sample 100 examples for each difficulty level (easy, medium, and hard), resulting in 300 test samples. For 2WikiMultiHopQA, we focus on 4 representative categories: bridge comparison, comparison, compositional, and inference. We randomly draw 75 examples from each category, leading to a total of 300 test samples.

**Results**   From Table 3, we can draw several conclusions. **(1)** Model-based CoT outperforms the Base method by a large margin across all three backbones. This indicates that decomposed semantics, when used as structured intermediate thoughts, facilitating the reasoning ability of the underlying LLM. **(2)** Although Model-based CoT underperforms compared to stronger baselines such as Reflexion and DoT-bank, ParamAgent achieves significant improvements over all state-of-the-art methods, even without incorporating a memory bank. This demonstrates the complementary advantages of semantic decomposition compared with reflective feedback, and highlights the effectiveness of combining them within a unified framework.

## 4.5 ADDITIONAL ANALYSIS

### 4.5.1 QUANTIFYING THE DIVERSITY OF REFLECTIONS

**Setup**   We study to what extent the reflection trajectories produced by ParamAgent differ from those of other baselines. To this end, we maintain the complete reflection history for each sample and embed each reflection using the OpenAI `text-embedding-3-small` model. We focus on the HumanEval dataset for this analysis. Concretely, for $N$ samples, and for each backbone LLM, we obtain a 2D tensor of shape $\mathbb{R}^{\sum_{i=1}^{N} n_i \times F}$, where $n_i$ denotes the number of reflection iterations for sample $i$, and $F$ is the embedding dimension. We then perform $k$-means clustering (Lloyd, 1982) over all reflections and apply the elbow method (Tibshirani et al., 2001) to determine the optimal clustering number $K^*$. This provides a quantitative measure of the semantic diversity of reflective feedback across methods. Additionally, for each $K$, we compute the silhouette score to evaluate clustering quality.

**Results**   As shown in Figure 2, ParamAgent achieves an optimal clustering number of $K^* = 39$, substantially larger than that of Reflexion, DoT and DoT-bank. This indicates that the reflective outputs of Reflexion lack diversity, whereas the parametric feedback of ParamAgent introduces significantly richer and more varied reflective signals. Moreover, across all clustering numbers $K$, the silhouette score of ParamAgent is consistently higher than those of competing methods. This suggests not only that ParamAgent generates more diverse reflections, but also that these reflections form more coherent semantic groups.

### 4.5.2 EFFECT OF REFLECTION FORMAT FROM $M_r$ ON PROGRAMMING TASKS

**Setup**   In this section, we conduct an ablation study on the content of the reflections produced by $M_r$ for programming tasks. As introduced in Section 3, for a given programming task $t$, $M_r$ samples two complementary components: (1) textual reflective feedback describing potential pitfalls, and (2)

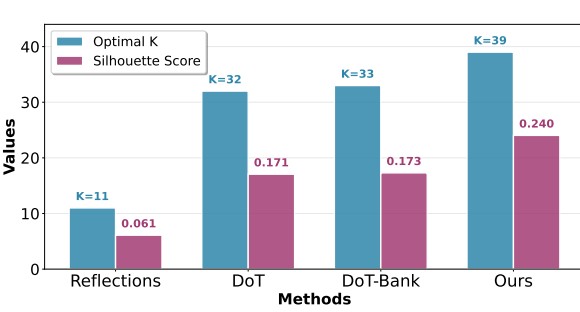 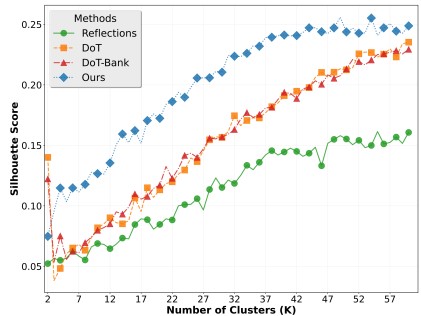

(a) Optimal clustering number $K^*$ across methods and the corresponding Silhouette scores.

(b) Silhouette score for every clustering number $K \in [2, 60]$.

Figure 2: Quantitative analysis of reflection diversity: (a) optimal clustering number $K^*$; (b) silhouette scores for different $K$.

code snippets implementing possible buggy solutions to illustrate failure modes. To analyze their contributions, we post-process the outputs from $M_r$ by removing one of the components. We then evaluate the resulting variants on the HumanEval dataset.

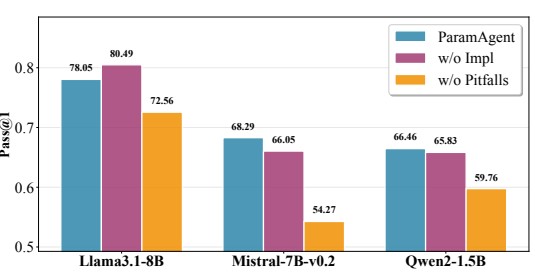 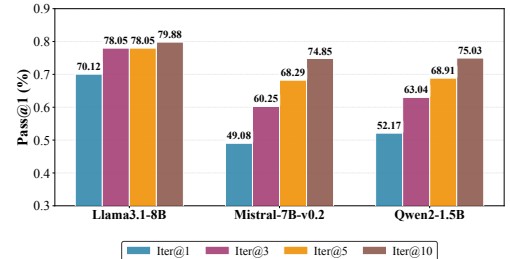

(a) Ablation study on the reflection format of $M_r$ on programming task.

(b) Ablation study on iteration numbers of Model-based Reflection on programming task.

Figure 3: Ablation studies.

**Results**   From Figure 3a, we can observe that: **(1)** The textual reflective feedback (w/o Impl) proves to be more important across different backbone LLMs. When only buggy code implementations are preserved without natural language descriptions, the performance sometimes even degrades, as observed in Llama-3.1-8B. **(2)** When only textual reflective feedback is kept, both Mistral-7B-v0.2 and Qwen2-1.5B show moderate drops in accuracy. This suggests that including buggy code implementations also provides useful complementary signals. Overall, these results indicate that outputting both natural language reflections and buggy code examples is a robust design choice.

### 4.5.3   EFFECT OF ITERATION NUMBER ON MODEL-BASED REFLECTION

**Setup**   In the main experiments, we fixed of iterations to 5 for fairness. Here we vary the iteration count of Model-based Reflection in $\{1, 3, 10\}$ to examine how diversity affect the model performance.

**Results**   As shown in Figure 3b, running only a single iteration yields subpar performance, indicating that providing one informative reflection ($T = 0.2$) is insufficient. Performance improves steadily as the iteration number increases significantly, even without an episodic memory buffer. This highlights that the diversity introduced by the parametric memory is crucial and effective for performance gains.

## 5   RELATED WORK

We discuss the most relevant work below, with additional related work in Appendix A.

**LLM Reasoning**    LLMs have demonstrated emergent abilities to perform multi-step reasoning when prompted appropriately. for instance, CoT prompting elicits the model to generate explicit intermediate reasoning steps and significantly improves performance on complex tasks (Wei et al., 2022). Self-Consistency (Wang et al., 2022) further improves CoT by sampling multiple reasoning paths and aggregating them via majority voting, which increases robustness. ReAct (Yao et al., 2023b) is a seminal approach that interleaves reasoning steps with actions (e.g., tool uses or environment queries) in an interactive decision-making loop, allowing the model to both "think" and "act" step-by-step. Other methods focus on iterative self-feedback, highlighting that reasoning is a process, not a one-shot (Madaan et al., 2023; Shinn et al., 2023). `ParamAgent` follows this iterative reasoning paradigm but avoids sophisticated search procedures.

**External Memory in LLMs**    Memory has become central for agents tackling multi-step reasoning (Zhang et al., 2024b). Short-term mechanisms such as Self-Refine (Madaan et al., 2023) use the model's own recent outputs as transient memory for iterative refinement, while Reflexion (Shinn et al., 2023) maintains episodic logs of errors and reflections to guide retries within a task. These approaches however, reset once a new problem begins. To address this limitation, external memory has been proposed to augment agentic reasoning (Borgeaud et al., 2022; Shi et al., 2023; Wang et al., 2023c; Shi et al., 2023; Zheng et al., 2023; Wang et al., 2024; Hu et al., 2024; Xu et al., 2025; Chhikara et al., 2025; Lingam et al., 2025). These methods mainly rely on non-parametric memory, either through textual logs or retrieval databases. By contrast, `ParamAgent` introduces a external parametric memory module $M_*$, which retrieves the knowledge from model-based sampler rather than recalling raw traces, enabling it to generate population-level knowledge that can be adapted to different domains.

**Cognitive Science Inspirations for Reasoning**    Cognitive science research shows that people manage complexity by chunking information into structured units, thereby reducing cognitive load and effectively increasing capacity for multi-step reasoning (Miller, 1956). Classic working memory models further posit a central executive that maintains and manipulates only a small set of active items, motivating reasoning procedures that keep a compact buffer of intermediate results (Baddeley, 2020). In parallel, problem solving is often modeled as hierarchical decomposition, where a complex goal is resolved by recursively addressing sub-goals and recombining their solutions (Simon, 2012). These insights motivate semantic decomposition in multi-hop QA: parse a query into compact units (e.g., entities, relations, constraints, answer type) and an explicit sequence of inference steps; solve sub-queries sequentially while maintaining a small active buffer of intermediates; then integrate the chunks to produce the final answer.

**Diversity in LLM Reasoning**    A key challenge in multi-step reasoning is avoiding redundant or myopic thought patterns. Recent works therefore encourage diversity of reasoning paths. One simple but effective approach is self-consistency decoding (Wang et al., 2022), which samples multiple independent chains-of-thought and then selects the final answer by majority vote. Beyond this, researchers have proposed methods to actively inject diversity into the reasoning process. For example, prompting the model with different personas or perspectives (e.g. "Think like a mathematician" vs. "Explain as a teacher") can yield varied solution paths (Naik et al., 2023). Another line of work trains LLMs to generate diverse solutions through specialized learning algorithms: Flow of Reasoning framework (Yu et al.) the generation of reasoning steps as a search problem and uses a GFlowNet-based (Bengio et al., 2023) fine-tuning to stochastically sample multiple high-reward reasoning trajectories, achieving greater coverage of the solution space. The recent DoT framework (Lingam et al., 2025) explicitly tackles the lack of exploration by producing non-redundant self-reflections to ensure each iteration explores new solution paths, rather than repeating past failures. Empirically, DoT shows that encouraging such diversity yields substantial gains on challenging reasoning tasks. `ParamAgent` adopts the principle of diversification in a notably simple way: by drawing each new reasoning attempt from a high-temperature parametric sampler (the memory module). This high-temperature sampling from the learned model-based memory introduces stochasticity that is easy to implement yet effective at covering different problem-solving trajectories, without needing elaborate persona prompts or complex search procedure.

## 6 CONCLUSIONS

We propose `ParamAgent`, a language agent framework that introduces a parametric module to move beyond instance-level reflection. By encoding cross-sample regularities into model parameters, `ParamAgent` internalizes global patterns and synthesizes population-level insights in a generative manner. Across programming, mathematical reasoning, and multi-hop QA, `ParamAgent` delivers substantial performance gains over state-of-the-art baselines while maintaining cost-effectiveness, highlighting the potential of parametric memory as a plug-in module for buidling language agents.

## 7 ETHIC STATEMENT

This work follows the ICLR Code of Ethics and Code of Conduct. We use only publicly available datasets under their respective licenses. No personally identifiable information is collected or generated. Our work does not produce social harm or pose immediate safety risks.

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

# Appendix

CONTENTS

## A   ADDITIONAL RELATED WORK

**Parametric Memory in LLM Reasoning**   Compared with textual memory, parametric memory remains under-explored in LLM agents. While textual stores dominate due to interpretability and ease of use, recent studies have also explored encoding memory directly into model parameters, thereby avoiding the length limitations of textual memory. Character-LLM (Shao et al., 2023) fine-tunes role-playing agents with character experiences to faithfully simulate personas. HuaTuo (Wang et al., 2023b) tunes LLaMA (Touvron et al., 2023) with Chinese medical knowledge to enhance clinical QA and instruction following. DoctorGLM (Xiong et al., 2023) develops a Chinese medical dialogue system, demonstrating that physician-style models can be obtained with moderate fine-tuning cost. Radiology-GPT (Liu et al., 2023) instruction-tunes on radiology corpora to outperform general LLMs on imaging-focused tasks. These approaches directly fine-tune the base LLM, yielding specialized models tailored to particular domains. By contrast, our framework keeps the base agent intact and fine-tunes an external parametric module that generates domain-specific reflective cues or semantic decompositions. This modular design allows the parametric module to serve as a plug-in component for different agents, while reducing the risk of catastrophic forgetting in the backbone. A detailed justification of this external design choice is provided in Appendix C. More details on parametric memory in agentic reasoning can be found in Zhang et al. (2024b).

## B   PSEUDOCODES FOR PARAMAGENT

In this section, we present pseudocode for `ParamAgent` in Algorithm 2. We also include pseudocodes for Model-based Reflection and Model-based CoT in Algorithm 3 and `ParamAgent-plus` in Algorithm 4 for clarity.

## C   RATIONALE FOR EXTERNAL MODULE FINE-TUNING

We justify our choice to fine-tune an external parametric module rather than the base language agent.

---

**Algorithm 2** `ParamAgent`

---

**Require:** Dataset $\mathcal{D}$, Base LM $p_\theta$, Parametric Module $M_*$ with parameters $\psi$, Max iterations $T_{\max}$, Pass@k $K$

**Ensure:** Solutions for each task

1: **Initialize:** Episodic memory $\mathcal{M} \leftarrow \emptyset$
2: **for** each task $x \in \mathcal{D}$ **do**
3:      solved $\leftarrow$ False, $k \leftarrow 0$
4:      **while** $k < K$ **and not** solved **do**
5:          $t \leftarrow 1$, $y_{\mathrm{curr}} \leftarrow$ None
6:          **while** $t \leq T_{\max}$ **and not** solved **do**
7:              # Generate parametric insights for iteration $t$
8:              **if** $t = 1$ **then**
9:                  $T \leftarrow 0.2$          $\triangleright$ informative first-round sampling
10:             **else**
11:                 $T \leftarrow 1.0$          $\triangleright$ promote diversity thereafter
12:             **end if**
13:             **if** *task is coding/math* **then**
14:                 $r_{t-1}^g \sim p_\psi(\cdot \mid x; T)$          $\triangleright$ global reflection from $M_r$
15:             **else**          $\triangleright$ multi-hop QA
16:                 $Z \sim p_\psi(\cdot \mid x; T)$          $\triangleright$ semantic units from $M_p$
17:             **end if**
18:             # Combine parametric and episodic knowledge
19:             $r_{1:t-1} \leftarrow$ RETRIEVEREFLECTIONS$(\mathcal{M}, x)$          $\triangleright$ local (self) reflections up to $t-1$
20:             **if** *task is coding/math* **then**
21:                 $y_{\mathrm{curr}} \sim p_\theta(\cdot \mid x, r_{1:t-1}, r_{t-1}^g)$          $\triangleright$ global–local fusion
22:             **else**
23:                 $y_{\mathrm{curr}} \sim p_\theta(\cdot \mid x, Z, r_{1:t-1})$          $\triangleright$ semantic decomposition
24:             **end if**
25:             # Evaluate and update episodic memory
26:             (passed, feedback) $\leftarrow$ EVALUATE$(y_{\mathrm{curr}}, x)$
27:             **if** passed **then**
28:                 solved $\leftarrow$ True
29:             **else**
30:                 $r_t \leftarrow$ GENERATESELFREFLECTION$(y_{\mathrm{curr}}, \text{feedback})$
31:                 $\mathcal{M} \leftarrow \mathcal{M} \cup \{(x, r_t)\}$          $\triangleright$ store only self-reflections
32:             **end if**
33:             $t \leftarrow t + 1$
34:          **end while**
35:          $k \leftarrow k + 1$
36:      **end while**
37: **end for**

---

**Training objectives** Given an input $x$, the external module is trained to produce either a reflection $r$ or semantic units $Z = \{z_i\}_{i=1}^m$:

$$\max_\psi \ \mathbb{E}_{(x,r)\sim\mathcal{D}_r}\left[\log p_\psi(r \mid x)\right], \qquad \max_\psi \ \mathbb{E}_{(x,Z)\sim\mathcal{D}_Z}\left[\log p_\psi(Z \mid x)\right]. \tag{3}$$

At inference, the agent conditions on $r_k^g \sim p_\psi(\cdot \mid x)$ or on $Z \sim p_\psi(\cdot \mid x)$:

$$y \sim p_\theta(\cdot \mid x, r_{1:k}, r_k^g), \qquad y \sim p_\theta(\cdot \mid x, Z, r_{1:k}). \tag{4}$$

**Why not fine-tune the agent directly?** Directly fine-tuning the base LLM within the agent introduces the following challenges:

**(1) Distribution mismatch.** In practice, an agent generates reflections autoregressively as $p_\theta(r_k \mid x, r_{1:k-1})$. If we fine-tune the base model only on $p_\theta(r \mid x)$ without its own history, the training distribution no longer matches the inference distribution $p_\theta(r_k \mid x, r_{1:k-1})$. Bridging this gap would require sequence-level supervision and far more data due to the more complex distribution form.

---

**Algorithm 3** Model-based Reflection (CoT)

---

**Require:** Dataset $\mathcal{D}$, Base LM $p_\theta$, Parametric Module $M_*$ with parameters $\psi$, Max iterations $T_{\max}$, Pass@k $K$
**Ensure:** Solutions for each task
1: **for** each task $x \in \mathcal{D}$ **do**
2:     solved $\leftarrow$ False, $k \leftarrow 0$
3:     **while** $k < K$ **and not** solved **do**
4:        $t \leftarrow 1$
5:        **while** $t \leq T_{\max}$ **and not** solved **do**
6:           # Parametric guidance only (no episodic memory)
7:           $T \leftarrow \begin{cases} 0.2 & \text{if } t = 1 \\ 1.0 & \text{otherwise} \end{cases}$
8:           **if** *task is coding/math* **then**
9:              $r_{t-1}^g \sim p_\psi(\cdot \mid x; T)$             $\triangleright$ global reflection from $M_r$
10:          $y_t \sim p_\theta(\cdot \mid x, r_{t-1}^g)$
11:           **else**                      $\triangleright$ multi-hop QA
12:              $Z \sim p_\psi(\cdot \mid x; T)$            $\triangleright$ semantic units from $M_p$
13:              $y_t \sim p_\theta(\cdot \mid x, Z)$
14:           **end if**
15:           # Evaluate (no memory write)
16:           passed $\leftarrow$ EVALUATE$(y_t, x)$
17:           **if** passed **then**
18:              solved $\leftarrow$ True
19:           **end if**
20:           $t \leftarrow t + 1$
21:        **end while**
22:        $k \leftarrow k + 1$
23:     **end while**
24: **end for**

---

**(2) Capability interference.** The agent must also maintain $p_\theta(y \mid x, r_{1:k})$ to act (e.g., generate code or multi-hop answers). Pushing the same parameters toward a specialized model for reflection generation can interfere with this objective, degrading the agent's general problem-solving ability.

**Benefits of using external module** In the meantime, adopt an external LLM module for parametric knowledge introduces several advantages:

**(1) Simpler supervision.** Decoupling the base LLM and the external LLM model yields a simpler objective $p_\psi(r \mid x)$ or $p_\psi(Z \mid x)$ rather than the history-conditioned $p_\theta(r_k \mid x, r_{1:k-1})$, reducing modeling complexity and data requirements.

**(2) Modular knowledge forms.** The module can emit different forms of parametric knowledge (e.g., reflections $r$ for programming/math via $M_r$, semantic units $Z$ for multi-hop QA via $M_p$), complementing episodic self-reflection $r_{1:k}$ without altering the base agent.

**(3) Stability and reuse.** Keeping the base LLM in the agent fixed also avoids interference with $p_\theta(y \mid x, r_{1:k})$, mitigates catastrophic forgetting, and enables plug-in use across agents and backbones.

In conclusion, fine-tuning an external module rather than the base LLM offers a simpler training objective, preserves the general capabilities of the agent, and enables flexible plug-in usage across domains and backbones, which justifies this design choice.

# D    MORE EXPERIMENTAL DETAILS RESULTS

## D.1    DATASET STATISTICS

**Programming.** For programming tasks, we evaluate on HumanEval (Chen et al., 2021) and MBPP (Austin et al., 2021). HumanEval consists of 164 hand-written Python programming problems,

---

**Algorithm 4** `ParamAgent-plus`

---

**Require:** Dataset $\mathcal{D}$, Base LM $p_\theta$, Parametric Module $M_*$, Max iterations $T_{\max}$
1: **Init:** Episodic memory $\mathcal{M} \leftarrow \emptyset$, Memory bank $\mathcal{B} \leftarrow \emptyset$, Failed $\mathcal{F} \leftarrow \emptyset$
2: **Phase 1: Standard solving with memory banking**
3: **for** each task $x \in \mathcal{D}$ **do**
4:     **for** $t = 1$ to $T_{\max}$ **or until solved do**
5:         $r^g_{t-1} \sim p_\psi(\cdot \mid x)$ with $T=0.2$ if $t=1$ else $T=1.0$         $\triangleright$ Parametric insight
6:         $r_{1:t-1} \leftarrow \text{Retrieve}(\mathcal{M}, x);$     $y \sim p_\theta(\cdot \mid x, r_{1:t-1}, r^g_{t-1})$
7:         **if** Evaluate$(y, x)$ passes **then**
8:             Store $(x, y, r^g_{t-1})$ in $\mathcal{B}$; mark solved; **break**
9:         **else**
10:            $r_t \leftarrow \text{Reflect}(y);$    $\mathcal{M} \leftarrow \mathcal{M} \cup \{(x, r_t)\}$         $\triangleright$ Update episodic
11:         **end if**
12:     **end for**
13:     **if** not solved **then** $\mathcal{F} \leftarrow \mathcal{F} \cup \{x\}$
14:     **end if**
15: **end for**
16: **Phase 2: Memory-augmented reattempt**
17: **for** each $x \in \mathcal{F}$ **do**
18:     $\mathcal{T} \leftarrow \text{RetrieveTopK}(\mathcal{B}, x);$    $x_{\text{aug}} \leftarrow \text{Augment}(x, \mathcal{T})$
19:     **for** $t = 1$ to $T_{\max}$ **or until solved do**
20:         $r^g_{t-1} \leftarrow \text{Extract}(\mathcal{T})$ **or** $p_\psi(\cdot \mid x_{\text{aug}})$       $\triangleright$ Reuse or generate (same $T$ rule as above)
21:         $r_{1:t-1} \leftarrow \text{Retrieve}(\mathcal{M}, x) + \text{RetrieveByReflection}(\mathcal{B})$
22:         $y \sim p_\theta(\cdot \mid x_{\text{aug}}, r_{1:t-1}, r^g_{t-1});$    Evaluate and update $\mathcal{M}, \mathcal{B}$
23:     **end for**
24: **end for**

---

each accompanied by hidden unit tests and a small number of visible test cases. We additionally consider MBPP, which provides 974 crowd-sourced Python problems; following prior work, we use the 397 problems from the filtered evaluation split.

**Math.** For mathematical reasoning, we adopt the MATH dataset (Hendrycks et al., 2021b), which contains competition-style math problems spanning seven subjects including Algebra, Geometry, Number Theory, Counting and Probability, and Precalculus. We randomly sample a balanced subset across categories for evaluation.

**Multi-hop QA.** For multi-hop question answering, we use HotpotQA (Yang et al., 2018) and 2WikiMultiHopQA (Ho et al., 2020). In HotpotQA, we stratify by difficulty level and randomly sample 100 examples from each category (easy, medium, hard), yielding a total of 300 evaluation samples. For 2WikiMultiHopQA, we stratify by question type and randomly sample 75 examples from each of four categories (bridge comparison, comparison, compositional, inference), again yielding 300 samples in total. These stratified subsets ensure balanced evaluation across different reasoning styles.

Table 4: Datasets used for Programming, Math, and Multi-hop QA tasks.

| Task Type | Dataset Name | Size | Metric |
|---|---|---|---|
| Programming | HumanEval | 164 problems, $\sim$3 visible test cases/problem | Pass@1 |
| Programming | MBPP | 397 sampled problems | Pass@1 |
| Math | MATH | 278 sampled problems across 7 subjects | 0-1 Acc |
| Multi-hop QA | HotpotQA | 300 sampled problems (100 per difficulty) | 0-1 Acc |
| Multi-hop QA | 2WikiMultiHopQA | 300 sampled problems (75 per type) | 0-1 Acc |

## D.2 FINETUNING THE PARAMETRIC MODULE

**Programming** For programming tasks, we curate a dataset by sampling 4000 coding problems from the APP dataset (Hendrycks et al., 2021a) at introductory level. In addition, we synthesize 4200 problems using GPT-4o-mini, covering a diverse range of programming domains. The code templates and prompt used for data generation are provided in Figure 4. For each problem, GPT-4o-mini is further asked to produce potential mistakes along with buggy implementations. This yields a dataset of reflective signals and corresponding erroneous code examples. We then finetune LLaMA-3.1-8B with LoRA on this dataset to obtain the programming-specific parametric module $M_r$.

**Math** For mathematical reasoning, we leverage the MATH training set (Hendrycks et al., 2021b). From each subject area, we randomly sample 800 problems and adopt the same pipeline as in programming: GPT-4o-mini is prompted to produce reflective feedback and buggy derivations for each sampled problem. The resulting dataset is used to LoRA-finetune LLaMA-3.1-8B to instantiate $M_r$ for math reasoning.

**Multi-hop QA** For multi-hop QA, we randomly sample 10000 instances from the HotpotQA (Yang et al., 2018) and 2WikiMultiHopQA (Ho et al., 2020) training sets respectively. GPT-4o-mini is prompted to output structured semantic units (e.g., entities, relations, constraints, answer types, and sub-questions) for each example. We then apply LoRA finetuning to LLaMA-3.1-8B on this dataset to build the parametric module $M_p$.

Across all domains, during dataset construction we provide one carefully designed demonstration example in the prompt to GPT-4o-mini. This ensures that the generated outputs (reflective feedback, buggy code, or semantic units) adhere to the required format, making the synthetic supervision more reliable.

## D.3 MORE IMPLEMENTATION DETAILS

We use the TogetherAI API service[1] to access all backbone models in our experiments. Specifically, we call the following model identifiers in implementation:

- `meta-llama/Meta-Llama-3.1-8B-Instruct-Turbo`
- `mistralai/Mistral-7B-Instruct-v0.2`
- `arize-ai/qwen-2-1.5b-instruct`

In Section D.4, we use 70B scale LLMs in our framework, the model identifiers are:

- `meta-llama/Meta-Llama-3.1-70B-Instruct-Turbo`
- `Qwen/Qwen2.5-72B-Instruct-Turbo`

All experiments are implemented in PyTorch (Paszke et al., 2019).

## D.4 HOW DOES PARAMAGENT PERFORM WITH STRONGER BASE LLMS?

We further study the performance of `ParamAgent` when paired with stronger base models of around 70B parameters. Specifically, we use Llama-3.1-70B and Qwen2.5-72B-Instruct as the underlying LLMs, while keeping the parametric module fixed as Llama-3.1-8B. We evaluate on HumanEval for programming and HotpotQA for multi-hop QA. The results are reported in Table 5 and Table 6 respectively.

**Results.** Across tasks, `ParamAgent` achieves performance that is on par with, or even surpasses, state-of-the-art baselines. Moreover, `ParamAgent-plus` consistently outperforms the best baseline methods by a large margin, highlighting the effectiveness of the parametric module. It is worth noting that our parametric module itself is only an 8B model, yet it integrates effectively with base LLMs as large as 70B. This demonstrates the strong potential of our approach when scaled further.

---

[1] https://www.together.ai

```
1   CATEGORIES = [
2       # Core Text & Parsing
3       "String Manipulation",
4       "Regular Expression Parsing",
5       "Natural-Language Tokenisation",
6       "CSV / JSON Parsing",
7       "URL / URI Parsing",
8       "Text Justification / Word-Wrapping",
9       # Lists, Arrays, SEQ
10      "Array / List Algorithms",
11      "Two-Pointer / Sliding-Window",
12      "Sorting & Searching",
13      "Statistical Summary of Sequences",
14      # Maths & Numbers
15      "Elementary Arithmetic / Algebra",
16      "Number Theory & Divisibility",
17      "Bitwise Operations",
18      "Combinatorics & Counting",
19      "Probability / Statistics",
20      # Data-Structures
21      "Hash / Set / Dict Operations",
22      "Stack / Queue Simulation",
23      "Linked-List Manipulation",
24      "Matrix Operations",
25      "Heap / Priority Queue Operations",
26      "Trie / Prefix-Tree",
27      # Graphs & Trees
28      "Graph / Tree Traversal",
29      "Binary Search Trees",
30      "Dynamic Programming",
31      "Recursion / Backtracking",
32      "Union-Find / Disjoint Set",
33      # Geometry / Coordinates
34      "Geometry & Coordinate Computation",
35      # Dates / Times / Calendars
36      "Date & Time Calculations",
37      # Miscellaneous Practical
38      "File & Path Utilities",
39      "Data-Type Conversion & Formatting",
40      "Cipher / Encoding",
41      "Simulation / Game Logic",
42      "Misc Small-Scale Algorithms"
43  ]
```

Figure 4: Schema of categories for synthesizing programming tasks used in our parametric module construction.

Table 5: Performance on HumanEval. **Bold** denotes the best result, and underline marks the second best. ↑ and ↓ indicate absolute change relative to the Base method. For clarity, the prompt token usage of the Base method is normalized to 1.

| Dataset | Method | Llama-3.1-70B-Instruct | | Qwen2.5-72B-Instruct | |
|---------|--------|------------------------|--------------|----------------------|--------------|
| | | Pass@1 | #Prompt Tokens | Pass@1 | #Prompt Tokens |
| **HumanEval** | Base | 80.49 | 1.00 | 82.92 | 1.00 |
| | Model-based Reflection | 87.80 ↑7.31 | 6.39 | 89.64 ↑6.72 | 3.48 |
| | Reflexion | 90.24 ↑9.75 | 4.31 | 88.41 ↑5.49 | 3.48 |
| | DoT | 90.85 ↑10.36 | 7.51 | 87.80 ↑4.88 | 6.05 |
| | DoT-bank | 92.68 ↑12.19 | 9.14 | 90.24 ↑7.32 | 8.17 |
| | ParamAgent | 92.07 ↑11.58 | 11.90 | 93.90 ↑10.98 | 8.93 |
| | ParamAgent-plus | **95.03** ↑14.54 | 19.47 | **95.12** ↑12.20 | 16.81 |

```
1   system_content = (
2       "You are an expert Python engineer crafting coding problems.\n"
3       "Follow this EXACT format:\n\n<template_example>\n\n"
4       "- Randomly pick ONE category from the list above.\n"
5       "- Output EXACTLY two lines:\n"
6       "    func_sign: <signature with colon>\n"
7       "    docstring: '<single-quoted string with \\n escapes>'\n"
8       "- Do NOT wrap in JSON or triple quotes.\n"
9       "- Avoid any collisions with past tasks.\n\n"
10  )
```

Figure 5: Prompt for synthesizing programming tasks

Table 6: Performance on HotpotQA dataset. **Bold** denotes the best result, and underline marks the second best. ↑ and ↓ indicate the absolute improvement or decrease relative to the Base method. For clarity, the prompt token usage of the Base method is normalized to 1.

| Dataset | Method | Llama-3.1-70B-Instruct | | Qwen2.5-72B-Instruct | |
|---|---|---|---|---|---|
| | | Acc | #Prompt Tokens | Acc | #Prompt Tokens |
| | Base | 70.00 | 1.00 | 73.33 | 1.00 |
| | Model-based CoT | 73.67 ↑3.67 | 1.43 | 74.10 ↑1.05 | 1.44 |
| | Reflexion | 82.33 ↑12.33 | 3.02 | 82.67 ↑9.34 | 2.81 |
| **HotpotQA** | DoT | 73.67 ↑3.67 | 3.43 | 80.67 ↑7.34 | 4.30 |
| | DoT-bank | 80.00 ↑10.00 | 5.24 | 82.33 ↑9.00 | 7.87 |
| | ParamAgent | 84.00 ↑14.00 | 7.70 | 81.00 ↑7.67 | 7.90 |
| | ParamAgent-plus | **89.67** ↑19.67 | 13.69 | **84.67** ↑11.34 | 15.43 |

Table 7: Token usage and cost on HumanEval and HotpotQA datasets with Llama3.1-8B as backbone LLM. Best and second-best metrics are in **bold** and underline respectively.

| Method | HumanEval | | | | HotpotQA | | | |
|---|---|---|---|---|---|---|---|---|
| | #Prompt Tokens | #Completion Tokens | Total Cost ($) | Pass@1 (%) | #Prompt Tokens | #Completion Tokens | Total Cost ($) | Acc (%) |
| Base | 37,463 | 13,506 | 0.00917 | 59.15 | 164,013 | 1,801 | 0.02985 | 57.67 |
| Model-based Reflection | 342,805 | 82,280 | 0.07652 | 78.05 | 236,548 | 1,212 | 0.04280 | 61.67 |
| Reflexion | 348,068 | 73,538 | 0.07589 | 76.22 | 703,192 | 68,612 | 0.13892 | 71.33 |
| DoT | 653,981 | 169,986 | 0.14831 | 72.56 | 1,164,812 | 106,806 | 0.22889 | 66.67 |
| DoT-bank | 926,047 | 233,016 | 0.20863 | 79.88 | 2,179,148 | 195,283 | 0.42740 | 72.00 |
| ParamAgent | 814,627 | 163,257 | 0.17602 | **82.93** | 3,649,598 | 128,010 | 0.67997 | **78.33** |

## D.5 COST ANALYSIS

Table 7 reports prompt/completion tokens and costs using Llama-3.1-8B. Costs are computed with TogetherAI pricing as of Aug 20, 2025 ($0.18 per million tokens). We can see that Model-based Reflection (CoT) is highly efficient, achieving strong accuracy with far fewer tokens than reflection-heavy methods like DoT-bank. By contrast, ParamAgent delivers the best results on both HumanEval and HotpotQA, at higher but still moderate cost, this highlights the advantages of incorporating various forms of memory modules.

## D.6 A CASE STUDY

We present a case study from the MBPP dataset, where both Reflexion and DoT fail to generate the correct implementation, while ParamAgent succeeds. To better understand this difference, we analyze the reflective history of all three methods and highlight the gists, as illustrated in Figure 6.

From the analysis, we observe that Reflexion and DoT often produce unhelpful sometimes even misleading reflections, which push the agent further away from the correct solution. In contrast, ParamAgent generates fewer such misleading reflections. We hypothesize that this advantage arises from the parametric knowledge encoded in $M_r$, which helps ParamAgent avoid unhelpful or error-prone reflective signals.

**Reflexion.**
**Problem:** "The implementation failed the test case where it tries to return a single element that is not a list. This issue arises because the ^ operator in the set difference operation returns a set, and then we explicitly convert it to a list..."
**Problem:** "The implementation failed the test where it is expected to find a difference in the lists because the list to check for difference is a subset of the original list... This would return the union of the two lists, not the actual difference."
**Problem:** "The implementation failed the test where the difference should be a single element because it uses the symmetric difference operation which is not what's intended here... When converting the set back to a list using the list function, the order of elements is lost."
**Problem:** "When converting the set back to a list using the list function, the order of elements is lost."

**DoT.**
**Problem:** "The implementation failed the test due to a mismatch in the type of the function return value and the expected output."
**Problem:** "The function implementation does not handle the case where the input lists are not of the same length or contain duplicate elements."
**Problem:** "The function implementation does not correctly calculate the symmetric difference between two lists."
**Problem:** "The implementation failed due to a mismatch in the type of the function return value and the expected output, but the proposed fix in the previous reflection is incomplete."
**Problem:** "The function implementation does not correctly handle the case where the input lists are not of the same length or contain duplicate elements..."
**Problem:** "The function implementation does not correctly remove duplicates from the input lists, leading to incorrect results."

**`ParamAgent.`**
**Problem:** "The function implementation does not handle the case of empty lists correctly."
**Fix:** "Add a check at the beginning of the function to return an empty list if either of the input lists is empty..."
**Problem:** "The function implementation does not preserve the original order of elements in the lists." **Fix:** "Instead of using set operations..., use a list comprehension to create the difference list..."
**Problem:** "The function implementation does not handle non-integer values in the lists correctly."
**Fix:** "Add a check at the beginning... ensure that both input lists are lists of integers..."
**Problem:** "The function implementation does not handle non-list types correctly." **Fix:** "Add a check... ensure that both input arguments are lists..."
**Problem:** "The function implementation does not handle duplicates correctly." **Fix:** "Use a list comprehension... and preserve the original order..."
**Problem:** "The function implementation does not handle the case where the lists are identical."
**Fix:** "Return an empty list if the two input lists are identical."

Figure 6: Selected reflections on the `mbpp_769_Diff` problem. **Red** = critical errors that directly invalidate the implementation; **Orange** = conceptual pitfalls or misleading reasoning patterns; **Green** = key insights leading to a working solution; **Gray** = background noise (irrelevant or stylistic).

### D.7 PROMPT TEMPLATES

We provide prompt templates used in `ParamAgent` across different domains. The 1-shot reflective example for programming tasks can be found in Figure 7, and the corresponding math reasoning template in Figure 8. For multi-hop QA, the semantic decomposition 1-shot example is shown in Figure 9.

Instruction templates for generating actions for the programming is shown in Figure 10, the math reasoning instruction in Figure 11, and the multi-hop QA instruction in Figure 12.

# E   LLM USAGE

We employed LLMs mainly for language polishing. All technical ideas, experimental analyses, claims, and writing decisions were conceived and verified by the authors.

```
1  [Function Signature]:
2  def has_close_elements(numbers: List[float], threshold: float) -> bool:
3      """Check if any two numbers in the list are closer than the threshold
           ."""
4
5  [Potential mistakes]:
6  1. **Empty or Single-Element Lists** must return `False`, not `True`.
7  2. **Duplicate Values** must be compared (difference 0), so never drop
       duplicates.
8  3. Always use **absolute difference** (`abs(a - b)`), not raw subtraction
       .
9  4. Use the correct **strictness** (`< threshold`, not `<=`).
10 5. Ensure you d o n t **exit too early** check  all distinct pairs.
11
12 [Flawed Implementations Illustrating Each Pitfall]:
13
14 def has_close_elements_v1(numbers: List[float], threshold: float) -> bool
       :
15     # BUG: returns True for empty or single-element lists
16     if len(numbers) < 2:
17         return True
18     for i in range(len(numbers)-1):
19         for j in range(i+1, len(numbers)):
20             if abs(numbers[i] - numbers[j]) < threshold:
21                 return True
22     return False
23
24 def has_close_elements_v2(numbers: List[float], threshold: float) -> bool
       :
25     # BUG: removes duplicates, so identical values never compared
26     numbers = sorted(set(numbers))
27     for i in range(len(numbers)-1):
28         if abs(numbers[i+1] - numbers[i]) < threshold:
29             return True
30     return False
31
32 def has_close_elements_v3(numbers: List[float], threshold: float) -> bool
       :
33     # BUG: uses raw subtraction instead of abs()
34     for i in range(len(numbers)-1):
35         for j in range(i+1, len(numbers)):
36             if (numbers[i] - numbers[j]) < threshold:
37                 return True
38     return False
39
40 def has_close_elements_v4(numbers: List[float], threshold: float) -> bool
       :
41     # BUG: uses <= instead of <, misclassifies exactly-threshold pairs
42     for i in range(len(numbers)-1):
43         for j in range(i+1, len(numbers)):
44             if abs(numbers[i] - numbers[j]) <= threshold:
45                 return True
46     return False
47
48 def has_close_elements_v5(numbers: List[float], threshold: float) -> bool
       :
49     # BUG: breaks out of outer loop too soon
50     ... (omit due to limited page)
51
52 END OF EXAMPLE
```

Figure 7: 1-shot example for reflective dataset construction for programming task.

**Question.** Circle $O$ is located on the coordinate plane with center at $(2, 3)$. One endpoint of a diameter is at $(-1, -1)$. What are the coordinates of the other endpoint of this diameter? Express your answer as an ordered pair.

**Pitfalls & Potential Mistakes**

1. **Confusing the center with an endpoint.** Assuming the center is an endpoint leads to an incorrect reflection point.

2. **Incorrect use of the midpoint formula.** Forgetting that the center is the midpoint of the diameter, or solving $(x + x_2)/2 = \text{center}_x$ incorrectly.

3. **Using the wrong coordinates for the midpoint.** Plugging endpoint coordinates in place of the center (or vice versa) yields the wrong unknowns.

4. **Arithmetic errors.** Sign or algebra mistakes when solving, e.g. $2 = (-1 + x)/2 \Rightarrow x = 3$ (incorrect) instead of $x = 5$.

5. **Switching $x$ and $y$.** Mixing $x$- and $y$-midpoint formulas, or using $x$ values to solve for $y$.

6. **Incorrect interpretation of the diameter.** Thinking the diameter extends in the same direction from the center; doubling the vector or reflecting in the wrong direction.

Figure 8: 1-shot example for reflective dataset construction in math reasoning.

**Example 1**
**Question.** Anatoly Maltsev and Valentin Turchin were both from Russia, which of the two is known for his work as a mathematician?

---

**Question Parsing and Intent Extraction**

**Key Components**

- **Entity A:** Anatoly Maltsev — mathematician/logician; contributions in mathematical logic and abstract algebra.

- **Entity B:** Valentin Turchin — computer scientist/philosopher; work in cybernetics and philosophy of science.

- **Implied Relationship:** Comparative inquiry: which individual is more closely associated with mathematics.

- **Answer Type Expected:** Person name (e.g., "Anatoly Maltsev").

- **Reasoning Type:** Comparative factual reasoning.

- **Required Background:** Biographical profiles or retrieved professional records.

---

**Inference Trace**

1. Retrieve factual data about Maltsev's and Turchin's primary academic domains.

2. Classify Maltsev as a mathematician (core contributions to mathematical logic).

3. Classify Turchin as mainly in cybernetics and philosophy.

4. Eliminate Turchin as the primary mathematician.

5. Conclude: **Anatoly Maltsev**.

---

**Disambiguation Note**
Nationality (Russia) does not help differentiate them.

Figure 9: 1-shot example used in `ParamAgent` for semantic decomposition dataset construction in multi-hop QA.

You are an AI Python assistant. You will be given some potential pitfalls and several flawed implementations for the coding challenge, as well as your previous implementation of a function, a series of unit-test results, and your self-reflection on your previous implementation. Try to avoid the errors from your previous implementation and the listed pitfalls.

**Instruction:** ALWAYS WRITE your full implementation (restate the function signature).

Figure 10: Instruction prompt used by `ParamAgent` to generate next-round solutions for programming tasks.

**You are revising your previous answer to a mathematics problem.**
You will receive:
(1) the original question,
(2) potential mistakes and pitfalls,
(3) your last answer, (4) feedback (Right or Wrong) explaining why that answer was unsatisfactory, and (5) your brief self-reflection on the mistake.

**Respond with:**
1. **Reasoning**: updated step-by-step thoughts.

2. **Answer**: the corrected final result.

**Formatting:** The final answer should be simplified to its simplest form, e.g., $25$, $25_{16}$, $\frac{1}{36}$, etc.

Figure 11: Instruction prompt used by `ParamAgent` to generate next-round solutions for math reasoning.

You are revising your previous answer to a multi-hop QA question.
You will receive:
(1) the original question,
(2) some key points, the underlying intent, and possible inference patterns that facilitate answering this question,
(3) your last answer,
(4) supporting context,
(5) feedback (Right or Wrong) explaining why that answer was unsatisfactory,
(6) your brief self-reflection on the mistake.

**Instruction:** Based on the inputs, produce a new single-phrase answer that resolves the error and fully answers the question. Output only the answer — no commentary, no code.

Figure 12: The prompt of `ParamAgent` to generate next-round answers for multi-hop QA tasks.

