# OpenReview forum: "ParamAgent: Language Agents with Parametric Knowledge"
_ICLR.cc/2026/Conference — ICLR 2026 Conference Withdrawn Submission_

### Official Review · Reviewer_p3C1 · 2025-10-16

**Soundness:** 2
**Presentation:** 2
**Contribution:** 1
**Rating:** 2
**Confidence:** 4

**Summary:**

This paper introduces ParamAgent, a language agent framework designed to overcome the limitations of instance-level reasoning and non-parametric memory. The core idea is to augment a base LLM with a domain-adaptive, external parametric module. This module is fine-tuned to internalize domain knowledge. The authors demonstrate the framework's flexibility by instantiating this module in two ways: (1) a reflection-oriented module (Mr) for programming and math tasks, which synthesizes global reflections on common pitfalls, and (2) a decomposition-oriented module (Mp) for multi-hop QA, which breaks down complex questions into structured semantic units.

The framework is shown to be compatible with different base LLMs and can be combined with existing memory mechanisms like episodic reflection and retrieval-based memory banks. Extensive experiments on programming (HumanEval, MBPP), math reasoning (MATH), and multi-hop QA (HotpotQA, 2WikiMultiHopQA) benchmarks show that ParamAgent and its variants consistently and significantly outperform state-of-the-art baselines, demonstrating the power of integrating parametric knowledge into agentic reasoning.

**Strengths:**

- The primary strength is the concept of a parametric knowledge module. This approach elegantly sidesteps the limitations of both purely instance-level reasoning (by incorporating global patterns) and non-parametric retrieval (by allowing the synthesis of novel, context-specific insights rather than just recalling past traces).

**Weaknesses:**

- Dependency on a "Teacher" Model: The parametric module M* is trained on data generated by GPT-4o-mini, a highly capable proprietary model. This means the method relies on knowledge distillation from a superior "teacher" model. It would be valuable to understand how much of the performance gain is due to the ParamAgent architecture itself versus the distilled knowledge from a stronger model. An ablation study where M* is trained on data generated by a weaker model (e.g., the base agent itself) would help isolate the architectural benefits.
- Static Parametric Memory: The core design of M* is that it is trained offline and remains static during inference. It introduces a significant limitation: the agent cannot generalize new experiences learned "on the fly" into its long-term parametric knowledge.
- Comparison to Simpler Alternative Frameworks: The paper overlooks comparisons to other established, and potentially simpler, paradigms for knowledge internalization. For example: (1) Direct Fine-tuning/Distillation: A straightforward alternative would be to collect successful solution trajectories from an agent (or a teacher model) and directly fine-tune the base LLM on them. (2) RL: Agentic reasoning tasks can be framed as sequential decision-making problems, making them amenable to RL methods like PPO/GRPO.
- Lack of Novelty: The proposed framework can be viewed as a sophisticated form of knowledge distillation. It distills domain-specific heuristics from a powerful teacher model (GPT-4o-mini) into a specialized module (M*) that generates a structured form of CoT-style guidance. This paradigm lacks the ability for true dynamic learning through environmental interaction. The agent cannot acquire genuinely new knowledge beyond what was pre-digested in M*'s offline training.
- Too Old Benchmarks and Base LLMs. There are many new, challenging benchmarks (e.g., SimpleQA, GPQA, SuperGPQA, LiveCodeBench) and SOTA LLMs (e.g., Qwen3 series). The old benchmarks and LLMs used in the paper make the results inconvicing.

**Questions:**

Please see Weakness

---

> ### Author Response · Authors · 2025-11-21
> **Reply to Reviewer p3C1**
>
> Thank you for your careful review and insightful comments! Please see below for our responses to your comments and concerns.
>
> ---
>
> > **W1: Dependency on a "Teacher" Model**
>
> We thank the reviewer for raising the concern regarding the use of GPT-4o-mini to generate training data for the parametric module $M_*$. To disentangle the effect of the teacher model from the contribution of our architecture, we performed additional experiments where $M_*$ is trained on data produced by **Llama-3.1-8B** itself, rather than GPT-4o-mini.
>
> For HumanEval, Llama-3.1-8B generated reflective feedback; for multi-hop QA, it generated semantic decompositions. The HumanEval results are shown below:
>
> **Table: Experimental results on HumanEval with Llama3.1-8B generated synthetic datasets**
> | Method                       | Llama-3.1-8B | Mistral-7B-v0.2 |
> |-----------------------------|-----------------:|---------------------:|
> | Base                        | 59.15%           | 32.93%              |
> | Model-based Reflection      | 78.05%           | 68.29%              |
> | Reflexion                   | 76.22%           | 51.22%              |
> | DoT                         | 73.17%           | 46.95%              |
> | DoT-bank                    | 79.56%           | 54.26%              |
> | ParamAgent (Llama-3.1-8B)   | 78.05%           | 65.85%              |
> | ParamAgent-plus (Llama-3.1-8B) | 86.59%        | 76.83%              |
> | ParamAgent (GPT-4o-mini)    | 82.93%           | 67.07%              |
>
>
> Two observations emerge:
>
> - **Architectural gains remain significant even without a strong teacher.** When trained on self-generated Llama-3.1-8B data, ParamAgent still improves over Reflexion and DoT, and ParamAgent-plus achieves **86.59%**, an **8.84%** improvement over DoT-bank. This shows that parametrizing reflections introduces a complementary form of diversity beyond prompt-level and retrieval-based diversity.
>
> - **A stronger teacher model further enhances performance.** ParamAgent trained on GPT-4o-mini data achieves **82.93%**, higher than the 78.05% obtained with Llama-3.1-8B supervision. As we have shown in Appendix D.6 in the draft, the reflection with weaker LLMs tend to be more misleading, therefore employing a stronger LLM like GPT-4o-mini may lead to a higher-quality reflective dataset, therefore bring addtional gains.
>
> A similar pattern holds for multi-hop QA:
>
> **Table: Experimental results on HotpotQA with Llama3.1-8B generated synthetic datasets**
> | Method                       | Llama-3.1-8B | Mistral-7B-v0.2 |
> |-----------------------------|-----------------:|---------------------:|
> | Base                        | 57.67%           | 45.00%              |
> | Model-based CoT             | 61.67%           | 54.33%              |
> | Reflexion                   | 71.33%           | 62.33%              |
> | DoT                         | 66.67%           | 58.33%              |
> | DoT-bank                    | 72.00%           | 66.33%              |
> | ParamAgent (Llama-3.1-8B)   | 76.33%           | 68.67%              |
> | ParamAgent-plus (Llama-3.1-8B) | 83.33%        | 81.00%              |
> | ParamAgent (GPT-4o-mini)    | 78.33%           | 69.67%              |
>
> Overall, these results confirm that (1) ParamAgent’s improvements are not solely dependent on a strong teacher model, the architectural design itself provides substantial gains; (2) while stronger supervision further enhances performance.

---

> ### Author Response · Authors · 2025-11-21
> **Reply to Reviewer p3C1 - Part2**
>
> > **W3: Knowledge Distillation and Environment Interaction
>
> We thank the reviewer for the comment.
>
> - **Success of paramAgent.** As shown in our additional experiments, even when $M_*$ is trained on data generated by the *same* base model (Llama-3.1-8B), ParamAgent and ParamAgent-plus still outperform Reflexion, DoT, and DoT-bank. This indicates that the main contribution is not distilling heuristics from a stronger teacher, but that the **parametrized module introduces an additional form of diversity**, complementary to prompt-level and retrieval-based diversity. Section 4.5.2 quantitatively confirms this effect, and Appendix D.5 in our draft provides a case study showing that the enriched diversity enables the agent’s reflections to identify the correct correction signals. In contrast, relying solely on Reflexion or DoT fails on some examples.
>
> - **Dynamic Learning.** We agree that $M_*$ alone (as in model-based reflection or model-based CoT) does not perform dynamic environmental interaction, these ablations were designed to isolate this factor. However, when $M_*$ is used *together* with self-reflection (ParamAgent) or cross-sample trajectories (ParamAgent-plus), its outputs **influence the agent’s subsequent actions**, indirectly influencing the iterative interaction dynamics. Thus, while $M_*$ itself is indeed offline, its integration still affects the agent’s behavior during multi-round interaction.
>
> > **W4: Old Benchmarks and  Base LLMs**
>
> We have added experiments using Qwen3-Next-70B-Instruct, a recent high-performance open-source model, and evaluated on LiveCodeBench (we subsample 50 problems for each difficulty level, leading to 150 questions in total) and HumanEval datasets. ParamAgent continues to yield consistent improvements.
>
> **Table: Qwen3-70B Results (LiveCodeBench + HumanEval)**
> | Method                 | LiveCodeBench | HumanEval |
> |------------------------|------------------------|---------------------|
> | Simple                 | 52.00%                 | 90.24%             |
> | Reflexion              | 62.00%                 | 96.34%             |
> | Model-based Reflection | 56.00%                 | 93.90%             |
> | DoT                    | 60.67%                 | 96.34%             |
> | DoT_bank               | 62.00%                 | 96.95%             |
> | ParamAgent             | 61.33%                 | 97.56%
> | ParamAgent-plus    | 63.33%             | 98.17%         |
>
> These results demonstrate that ParamAgent remains competitive under modern LLMs and recent, harder benchmarks.
>
> As SimpleQA and GPQA are closed-book factuality benchmarks, in contrast, our work targets multi-step, context-dependent reasoning, therefore these datasets do not align with our objective. To address the reviewer's concern, we add experiments on HotpotQA using Qwen3-Next-70B-Instruct, as shown below.
>
> **Table: Qwen3-70B Results (HotpotQA)**
> | Method                | Accuracy |
> |-----------------------|----------|
> | Simple                | 71.33%   |
> | Reflexion             | 82.33%   |
> | Model-based Reflection | 76.33% |
> | DoT                  | 79.67%   |
> | DoT-bank             | 83.33%   |
> | ParamAgent           | 79.67%   |
> | ParamAgent-plus   | 85.00% |
>
> We can observe that the parametric module also improves the performance of Qwen3-70B, even though the parametric module itself is only an 8B-scale model.
>
> ---
>
> We sincerely thank the reviewer for the careful review and thoughtful comment. We hope that our responses have addressed all concerns. Please do not hesitate to let us know if there are any remaining concerns.

---

> > ### Comment · Reviewer_p3C1 · 2025-11-24
> >
> > Thank you for the response. I still tend to maintain my original score because: (1) most improvements are marginal and rely on strong models; (2) the methodology lacks novelty

---

> ### Author Response · Authors · 2025-11-26
> **Reply to Reviewer p3C1**
>
> Thank you for your continued engagement. We respectfully address both concerns below.
>
> > ###  **"Most improvements are marginal and rely on strong models"**
>
> We have conducted experiments where the synthetic dataset is generated by the **same base LLM** (Llama-3.1-8B), not a stronger teacher model. To ensure fair comparison, we focus on:
>
> - ParamAgent vs. DoT: The only difference is the addition of our parametric module.
> - ParamAgent-plus vs. DoT-bank: Similarly, only differs by our parametric module.
>
> **Table: Results with Llama-3.1-8B generated data**
> | Task | ParamAgent vs. DoT | ParamAgent-plus vs. DoT-bank |
> |------|-------------------|------------------------------|
> | HumanEval | 78.05% vs. 73.17% (**+6.67%**) | 86.59% vs. 79.56% (**+8.84%**) |
> | HotpotQA | 76.33% vs. 66.67% (**+14.49%**) | 83.33% vs. 72.00% (**+15.74%**) |
>
> These results demonstrate that: (1) our method does **not** rely on strong teacher models to achieve effectiveness; (2) the improvements are not marginal (up to +15.74%).
>
> > ### **"The methodology lacks novelty"**
>
> We respectfully clarify our contributions and technical novelty:
>
> 1. **Novel Finding on Reflexion**: Beyond the known issue of lacking diversity, we discover that many reflections generated by weaker LLMs are **misleading** and fail to capture the core problem (Appendix D.6). This motivates our parametric approach.
>
> 2. **Fundamentally Different Diversity Mechanism**: Unlike prompt-level diversity or embedding-based retrieval diversity, our parametric memory captures cross-sample correlations through **training dynamics** and synthesize insights in a **generative** way. Our empirical analysis shows this introduces a **complementary form of diversity** that stacks on top of existing mechanisms.
>
> 3. **Flexible Synthetic Data Strategy**: While using the same base LLM yields significant gains, stronger LLMs can further enhance performance. Our analysis shows stronger LLMs produce more accurate reflections, especially benefiting smaller models, which is able to achieve performance on par with stronger LLMs. However, we emphasize that **even without a stronger teacher, our method remains effective**.
>
> 4. **Zero-shot Generalization**: We train the parametric module on one dataset and evaluate on others without further finetuning (e.g., for multi-hop QA, train on HotpotQA, evaluate on 2WikiMultiHopQA; for programming, we also train on synthetic dataset, directly perform evaluation on HumanEval, MBPP, and LiveCodeBench). Across all backbones, ParamAgent vs. DoT shows **significant improvements even in zero-shot settings**, demonstrating the zero-shot generalizability of our approach.
>
> ---
>
> We hope that these additional clarifications adequately address your concerns. Should any questions remain, we welcome further discussion and would be happy to provide additional clarification.

---

### Official Review · Reviewer_zuPE · 2025-10-26

**Soundness:** 2
**Presentation:** 3
**Contribution:** 2
**Rating:** 2
**Confidence:** 4

**Summary:**

The paper proposes ParamAgent, a framework that coordinates multiple agents for reasoning tasks such as mathematics, programming, and multi-hop question answering. Specifically, the framework involves a main reasoning agent, an LLM that provides reflections, and another LLM that performs question decomposition for multi-hop QA. Empirical results suggest that ParamAgent outperforms both prompting-based and retrieval-based reflection methods, as well as a model-based reflection variant used as an ablation.

My concerns are twofold:

1. Limited technical novelty.
 ParamAgent appears to be an ensemble of existing ideas rather than a conceptually new framework. It builds upon well-known paradigms such as multi-agent collaboration [1], reflexion [2], and question decomposition [3,4]. From the presentation, the authors seem to emphasize the parameterization of external reflection memory as their main contribution (hence the name ParamAgent). However, according to Algorithm 1, the method still relies on self-reflection and external memory to store the self-generated reflections and uses a retriever to query relevant ones. This design choice raises questions about the true motivation and distinctiveness of the work. The authors criticize prior self-reflection methods for their lack of population-level insights and retrieval-based methods for shallow interaction, yet ParamAgent itself remains heavily dependent on self-reflection and retrieval. Moreover, removing these components causes a notable drop in performance (as shown by the comparison between ParamAgent and the model-based variant). Taken together, the technical contribution seems incremental.


2. Weak empirical justification.
 The experimental setup does not convincingly demonstrate the practical usefulness of ParamAgent. The base models used in the experiments are considerably weaker than current state-of-the-art systems. In practice, users are more likely to employ stronger open-source models such as Qwen3 or DeepSeek-r1. Under such circumstances, the relevance of the reported improvements is unclear. The comparison with Reflexion may also be unfair, since that method was primarily verified with large-scale closed-source LMs. Finally, it would be informative to include a simpler baseline—for example, removing Line 5 from Algorithm 1—to clarify whether ParamAgent offers substantial benefits over minimal modifications to the workflow.

[1] Qingyu Wu et al., AutoGen: Enabling Next-Gen LLM Applications via Multi-Agent Conversation

[2] Noah Shinn et al., Reflexion: Language Agents with Verbal Reinforcement Learning

[3] Danny Zhou et al., Least-to-Most Prompting Enables Complex Reasoning in Large Language Models

[4] Jian Guan et al., AMOR: A Recipe for Building Adaptable Modular Knowledge Agents Through Process Feedback

**Strengths:**

1. A multi-agent system for reasoning
2. the system is verified with multiple types of reasoning tasks

**Weaknesses:**

see my comments in Summary

**Questions:**

N/A

---

> ### Author Response · Authors · 2025-11-21
> **Reply to Reviewer zuPE**
>
> Thank you for your careful review and insightful comments. Please see below for our responses to your comments and concerns.
>
> ---
>
> > **W1: Relation to Prior Multi-Agent Frameworks**
>
> - **Relations with multi-agent system.** We acknowledge that ParamAgent can be viewed as a form of multi-agent system. However, multi-agent frameworks encompass a wide range of technical realizations, coordination via debate, task decomposition, voting, self-consistency, etc. Our design differs from prior forms of multi-agent system: instead of introducing new subtasks or agent debates, we propose a parametrized module that enhances the agent’s reasoning ability by injecting **a new source of information diversity** during reflection. This is fundamentally different from traditional agent collaboration, which typically focuses on distributing subtasks, debate with multiple agents, or combining independent reasoning processes.
>
> - **Clarifying the role of self-reflection and retrieval.** We agree with the reviewer that parts of the introduction may have unintentionally overstated the limitations of prior reflection systems. We have revised the draft to clarify that our goal is not to replace self-reflection or retrieval; rather, the parametrized module is designed to **work jointly with these components**. $M_*$ adds an additional layer of diversity beyond:
>
>   - prompt-level diversity (as in DoT), and
>   - retrieval-based diversity (as in previous studies that use external memoty for text retrieval).
>
> This complementary diversity enhances agent reflection, rather than serving as an alternative to replace the reflection mechanism. The empirical results in Section 4.5.2 in our draft also support this argument.

---

> ### Author Response · Authors · 2025-11-21
> **Reply to Reviewer zuPE - Part2**
>
> > **W2: Empirical Justification**
>
> We thank the reviewer for the concerns and address them point by point.
>
> ### **Practical relevance under stronger modern LLMs.**
>
> We agree that users increasingly rely on stronger open-source models. To directly address this, we have added new experiments using Qwen3-Next-80B-Instruct, a recent and competitive open-source model, along with evaluations on LiveCodeBench, a substantially more challenging benchmark than HumanEval. We can see that ParamAgent continues to provide consistent improvements.
>
> **Table: Qwen3-80B Results on LiveCodeBench and HumanEval**
> | Method                 | LiveCodeBench | HumanEval |
> |------------------------|------------------------|---------------------|
> | Simple                 | 52.00%                 | 90.24%             |
> | Reflexion              | 62.00%                 | 96.34%             |
> | Model-based Reflection | 56.00%                 | 93.90%             |
> | DoT                    | 60.67%                 | 96.34%             |
> | DoT_bank               | 62.00%                 | 96.95%             |
> | ParamAgent             | 61.33%                 | 97.56%             |
> | ParamAgent-plus    | 63.33%             | 98.17%         |
>
> We further evaluated HotpotQA using Qwen3-80B model:
>
> **Table: Qwen3-80B Results on HotpotQA**
> | Method                | Accuracy |
> |-----------------------|----------|
> | Simple                | 71.33%   |
> | Reflexion             | 82.33%   |
> | Model-based Reflection | 76.33% |
> | DoT                  | 79.67%   |
> | DoT-bank             | 83.33%   |
> | ParamAgent           | 79.67%   |
> | ParamAgent-plus   | 85.00% |
>
> These results also show that the parametrized semantic units consistently improves Qwen3-70B performance, despite the parametric module being only an **8B-scale** model.
>
> ### **Comparison with Reflexion on closed-source LLMs.**
>
> We thank the reviewer for raising this point. ParamAgent relies on finetuning a parametric module, and for closed-source LLMs we cannot directly finetune the underlying model to fully validate the performance of the framework. To assess whether our method is still effective under identical model capacity, we conduct controlled experiments where both the auxiliary datasets and the agent itself use the *same* LLM.
>
> Concretely, we use Llama-3.1-8B to generate the reflective or semantic-decomposition datasets and also evaluate the agent with Llama-3.1-8B. The results are shown below:
>
> **Table: Experimental result using Llama-3.1-8B to build the auxialary datasets**
> | Method            | HumanEval | HotpotQA |
> |-------------------|---------------:|--------------:|
> | Base              | 59.15%         | 57.67%        |
> | Model-based Reflection | 78.05%    | 61.67%        |
> | Reflexion         | 76.22%         | 71.33%        |
> | DoT               | 73.17%         | 66.67%        |
> | DoT-bank          | 79.56%         | 72.00%        |
> | ParamAgent        | 78.05%         | 76.33%        |
> | ParamAgent-plus   | 86.59%         | 83.33%        |
>
>
> Even when using the *same* LLM for both dataset generation and agent execution, thus isolating the effect of a stronger LLM for knowledge transfer, ParamAgent still provides substantial improvements.
>
> Therefore, although we cannot finetune closed-source LLMs, these controlled results indicate that applying ParamAgent on top of a finetunable closed-source model (if exposed) should likewise **yield meaningful gains**.
>
>
> ### **Including a simpler baseline (removing Line 5 of Algorithm 1).**
>
> If Line 5 is removed, the workflow effectively reverts to **standard Reflexion**, since the agent no longer incorporates the parametric module’s outputs during reflection. As shown in all experimental settings, ParamAgent and ParamAgent-plus consistently outperform Reflexion, confirming that the parametric module provides additional value to the reflection process.
>
> ---
>
> We sincerely thank the reviewer for the careful review. We hope that our responses have addressed all concerns. Please do not hesitate to let us know if there are any remaining concerns.

---

### Official Review · Reviewer_2AZY · 2025-10-30

**Soundness:** 2
**Presentation:** 2
**Contribution:** 2
**Rating:** 4
**Confidence:** 3

**Summary:**

This paper addresses two key limitations of existing Large Language Model (LLM)-based agent frameworks: their confinement to per-instance reasoning (overlooking cross-task transferable patterns) and reliance on nonparametric external memory (capturing only shallow cross-instance interactions). To solve these issues, the authors propose ParamAgent, a language agent framework that integrates a domain-adaptive parametric module (M*) to internalize cross-sample knowledge into model parameters.

**Strengths:**

1.	ParamAgent addresses critical gaps in existing agents by using parametric memory to internalize deep cross-sample regularities—avoiding the shallowness of nonparametric memory (e.g., retrieval logs) and the inefficiency of per-instance reasoning.
2.	The modular design of  M_r and M_p enables customization for distinct tasks (programming/math vs. multi-hop QA), and the framework works with diverse base LLMs (from 1.5B to 70B scales), enhancing its practical applicability.
3.	Rigorous Experimental Design: Evaluations cover three distinct, challenging tasks with well-known benchmarks (HumanEval, MATH, HotpotQA) and multiple base models, ensuring generalizability. Ablation studies (e.g., reflection format, iteration count) and extended tests with 70B-scale LLMs provide deep insights into the framework’s behavior.

**Weaknesses:**

1. The authors should have clearly explained how M* works, as this is the core of the paper. However, it is placed in the appendix algorithm-3 section. Besides the algorithm, there should be a framework to illustrate how the proposed method works.

2. The paper relies on synthetic data generated by GPT-4o-mini to train M*, but provides insufficient details on: Whether the framework’s performance is sensitive to the size or diversity of the synthetic dataset (e.g., would smaller datasets degrade M* effectiveness?).

3. The paper claims “distribution mismatch” and “general capability degradation” of fine-tuning the base LLM, does this occurs during fine-tuning the M*?

4. What’s the difference between the paramagent and multi-agent?

**Questions:**

The same as weaknesses:


1. The authors should have clearly explained how M* works, as this is the core of the paper. However, it is placed in the appendix algorithm-3 section. Besides the algorithm, there should be a framework to illustrate how the proposed method works.

2. The paper relies on synthetic data generated by GPT-4o-mini to train M*, but provides insufficient details on: Whether the framework’s performance is sensitive to the size or diversity of the synthetic dataset (e.g., would smaller datasets degrade M* effectiveness?).

3. The paper claims “distribution mismatch” and “general capability degradation” of fine-tuning the base LLM, does this occurs during fine-tuning the M*?

4. What’s the difference between the paramagent and multi-agent?

---

> ### Author Response · Authors · 2025-11-21
> **Reply to Reviewer 2AZY**
>
> Thank you for your careful review and insightful comments! Please see below for our responses to your comments and concerns.
>
> ---
>
> > **W1: Clarity of How the Parametric Module $M_*$ Works**
>
> We thank the reviewer for the comment. As shown in Algorithm 1 (Lines 177–179) and line 160-165 in the main text, the use of $M_*$ is intentionally simple:  during each reflection step, the agent samples a reflection from $M_*$ and concatenates it with its self-generated reflection. This combined reflective signal is then given as input to the agent to generate the solution.
>
> > **W2: Sensitivity to the Size and Diversity of Synthetic Data**
>
> We performed an additional experiment where we reduced the synthetic dataset by half, removing the APP subset for the coding tasks. We then retrained $M\_r$ and evaluated ParamAgent on HumanEval using both Llama-3.1-8B and Mistral-7B-v0.2.
>
> The results are shown below:
>
> **Table: The impact of size of synthetic dataset**
> | Method                     | Llama-3.1-8B | Mistral-7B-v0.2 |
> |---------------------------|------------------:|----------------------:|
> | Base                      | 59.15%            | 32.93%               |
> | Model-based Reflection    | 78.05%            | 68.29%               |
> | Reflexion                 | 76.22%            | 51.22%               |
> | DoT                       | 73.17%            | 46.95%               |
> | DoT-bank                  | 79.56%            | 54.26%               |
> | ParamAgent (full dataset) | 82.93%            | 67.07%               |
> | ParamAgent (half dataset) | 81.70%            | 65.24%               |
>
>
> The performance of ParamAgent shows  a small drop when trained on half the data, indicating that $M\_r$ is not highly sensitive to dataset size. This is because $M\_r$ is trained via LoRA, updating only less than 1% of the model parameters, making the module sample-efficient.
>
> > **W3: Distribution Mismatch When Fine-Tuning $M_*$**
>
> We wish to clarify that fine-tuning the parametric module is different from fine-tuning the base agent LLM, as we discussed in Appendix C, $M_*$ is optimized only to fit the distribution
>
> $$r \sim p_\psi(r \mid x), \quad or \quad Z \sim p_\psi(Z \mid x),$$
>
> it specializes in generating reflections or semantic units conditioned solely on the input $x$. $M_*$ never generates actions of the form:
>
> $$y \sim p_\theta(\cdot \mid x, r_{1:k}, r_k^g),$$
>
> which are produced exclusively by the base agent model.
>
> Because $M\_*$ is trained only on the reflection distribution $p_\psi(r \mid x)$ or $p_\psi(Z \mid x)$, and does not participate in the action-generation distribution of the agent, there is no distribution mismatch, and the base model’s capabilities are not affected.
>
> >  **W4: Difference Between ParamAgent and a Multi-Agent System?**
>
> We thank the reviewer for this question. Multi-agent systems are a broad notion, including various techincal realizations, such as debate, collaboration, self-consistency, and task decomposition. ParamAgent can be viewed as a two-agent system, but its mechanism **is technically different from previous ones.**
>
> The auxiliary module $M_*$ is not another agent performing independent reasoning or debate. Instead, it is a specialized module, trained to internalize reflections or semantic units. Its role is to provide a **new source of diversity** that complements self-reflection and retrieval-based textual logs.
>
> In summary, we clarify that the **key contribution** of our work is the introduction of a parametrized module that directly augments the agent’s reasoning ability by injecting a new form of diversity in the reflective process in the agent. This is different from the current multi-agent system technically.
>
> ---
>
> We sincerely thank the reviewer for the efforts and constructive feedback. Please do not hesitate to let us know if there are any remaining concerns.

---

### Official Review · Reviewer_QdAz · 2025-10-31

**Soundness:** 3
**Presentation:** 4
**Contribution:** 3
**Rating:** 8
**Confidence:** 3

**Summary:**

This paper presents  new agentic framework called ParamAgent, where reflections for the task at hand are produced by a fine-tuned model along with self-reflections. Main goal of the paper is to gather global signals for a problem at hand with reflections from fine-tuned ParamAgent. Paper presents results on programming, math and multi-hop QA datasets with the proposed approach and show significant gains on all three tasks to show the effectiveness of the approach.  ParamAgent has two different forms parametric knowledge: 1) is global reflection useful for programming and reasoning tasks 2) semantic decomposition useful for QA tasks.

**Strengths:**

1. Paper is very well written and easy to follow.
2. Thorough experimentation to show the effectiveness of the proposed approach on three different types of benchmarks.

**Weaknesses:**

Model context slightly increases in all cases with more tokens provided across tasks.
Nice to have some experiments where multi-turn conversations are present, and how this works in those contexts?

**Questions:**

1. When you use model based reflection, do you retain the reflections at each step or is it similar to ParamAgent case where its only appended to the current turn? if its former then how does it fare if you use it only for current turn?
2. Most of these datasets are single query followed by models reasoning to arrive at the answer, how does this framework fare in cases of conversational or multi-turn interactions with the model, like Agentic use cases?

---

> ### Author Response · Authors · 2025-11-21
> **Reply to Reviewer QdAz**
>
> Thank you for your careful review and positive feedbacks! Please see below for our responses to your comments and concerns.
>
> ---
>
> > **W1: Multi-turn conversational and model context**
>
> We clarify that ParamAgent and ParamAgent-plus both operate in a multi-turn interaction setting, the agent repeatedly interacts with the environment, receives feedback, and updates its reasoning trajectory over multiple rounds. ParamAgent-plus leverages additional retrieved textual trajectories (thus more context) for reflection to introduce more diversity during these multi-turn updates. This also implies that longer context can benefit our framework with multi-turn interactions.
>
>
> > **Q1: Setting of model-based reflection**
>
> We clarify that in our model-based reflection setting, reflections are not accumulated across turns. When feedback from environment indicates failure, we simply resample a new reflection from $M_r$ and retry. Each reflection is sampled and used only for the current attempt.
>
> As shown in Figure 3(b) in the draft, even without any environment interaction, repeatedly sampling from $M_r$ already yields strong performance for the coding problem.
>
>
> > **Q2: multi-turn interactions of ParamAgent**
>
> Thank you for raising this point. Although the model-based variants operate without environment interaction, ParamAgent itself is fully agentic: it interacts with the environment over multiple turns and makes use of the reflective history accumulated during the reasoning process. Thus, while the model-based ablations are single-turn by design, the full ParamAgent framework naturally supports and benefits from multi-turn agent–environment interaction.
>
>
> ---
>
> We sincerely thank the reviewer once again for the careful review. We hope that our responses have addressed all the concerns. Please do not hesitate to let us know if there are any remaining concerns.

---

### Note · Authors · 2025-12-09

I have read and agree with the venue's withdrawal policy on behalf of myself and my co-authors.